# Phylogenetic Analysis and Characterization of Diguanylate Cyclase and Phosphodiesterase in Planktonic Filamentous Cyanobacterium *Arthrospira* sp.

**DOI:** 10.3390/ijms242015210

**Published:** 2023-10-16

**Authors:** Kang Wang, Wenjun Li, Hongli Cui, Song Qin

**Affiliations:** 1Yantai Institute of Coastal Zone Research, Chinese Academy of Sciences, Yantai 264003, China; wangkang211@mails.ucas.ac.cn (K.W.); wjli@yic.ac.cn (W.L.); hlcui@yic.ac.cn (H.C.); 2University of Chinese Academy of Sciences, Beijing 100049, China

**Keywords:** c-di-GMP, *Arthrospira*, diguanylate cyclase, phosphodiesterase, riboswitch

## Abstract

Cyclic di-GMP (c-di-GMP) is a second messenger of intracellular communication in bacterial species, which widely modulates diverse cellular processes. However, little is known about the c-di-GMP network in filamentous multicellular cyanobacteria. In this study, we preliminarily investigated the c-di-GMP turnover proteins in *Arthrospira* based on published protein data. Bioinformatics results indicate the presence of at least 149 potential turnover proteins in five *Arthrospira* subspecies. Some proteins are highly conserved in all tested *Arthrospira*, whereas others are specifically found only in certain subspecies. To further validate the protein catalytic activity, we constructed a riboswitch-based c-di-GMP expression assay system in *Escherichia coli* and confirmed that a GGDEF domain protein, Adc11, exhibits potential diguanylate cyclase activity. Moreover, we also evaluated a protein with a conserved HD-GYP domain, Ahd1, the expression of which significantly improved the swimming ability of *E. coli*. Enzyme-linked immunosorbent assay also showed that overexpression of Ahd1 reduced the intracellular concentration of c-di-GMP, which is presumed to exhibit phosphodiesterase activity. Notably, meta-analyses of transcriptomes suggest that Adc11 and Ahd1 are invariable. Overall, this work confirms the possible existence of a functional c-di-GMP network in *Arthrospira*, which will provide support for the revelation of the biological function of the c-di-GMP system in *Arthrospira*.

## 1. Introduction

Cyclic di-GMP (c-di-GMP) is a universal bacterial second messenger. Over the years, studies have shown its important role in regulating biofilm formation, polysaccharide synthesis, motility, virulence, intercellular communication, and interaction with host cells [1]. In addition, several studies have confirmed the potential role of c-di-GMP in cell differentiation in multicellular bacteria, such as *Anabaena* sp. PCC 7120 and *Streptomyces coelicolor*, but knowledge of this signal process is lacking [2,3,4].

Generally, diguanylate cyclases (DGCs) containing the GGDEF domain and c-di-GMP-specific phosphodiesterases (PDEs) containing the EAL or HD-GYP domain regulate c-di-GMP levels [1]. They are often highly redundant, can rapidly vanish from species within a genus, and possess above-average transmissibility [5]. The activity of DGCs and PDEs is controlled by a variety of N-terminal sensory domains, which can receive and respond to various signals, such as light and some small molecules [6]. C-di-GMP receptors (including various transcription regulators, riboswitches, etc.) are responsible for sensing and regulating changes in intracellular c-di-GMP concentrations and initiating downstream signaling or the expression of specific genes [7].

Studies on the c-di-GMP signaling system are vital for understanding signal perception and environmental adaptation in prokaryotes. However, there is a lack of reports on the c-di-GMP signaling networks in cyanobacteria, particularly in the multicellular filamentous cyanobacterium, *Arthrospira*. This economically important species is diverse in its filamentous morphology, which is affected by changes in environmental conditions, such as pH, temperature, and light intensity [8]. Regrettably, these underlying assumptions lack critical evidence. Therefore, due to its vital role in regulating cell morphology [2,3,4,9], and to enhance the knowledge of the signal system in filamentous cyanobacteria, investigating the c-di-GMP network of *Arthrospira* is necessary.

In this study, we analyzed the phylogenetic distribution of all predicted c-di-GMP turnover proteins in five *Arthrospira* subspecies and characterized two turnover proteins (Adc11 and Ahd1) with highly conserved distributions across the entire *Arthrospira* genus. The results suggest that Adc11 and Ahd1 are active DGCs and PDEs, respectively, and that their expression is unaffected by various physiological conditions, which supports the possibility that they may act as basic c-di-GMP-producing or -degrading genes in *Arthrospira* to maintain normal levels of c-di-GMP.

## 2. Results

### 2.1. The Phylogenetic Distribution of Predicted C-di-GMP Turnover Proteins

In 1998, two linear DNA fragments were isolated from *A. platensis* FS by Kojima et al. [10]. These fragments are homologous to the *A. platensis* genome, and bioinformatic results suggested that these two fragments might be associated with DGCs that catalyze the synthesis of c-di-GMP [10]. However, knowledge of the c-di-GMP network in *Arthrospira* is extremely limited to date, and there are almost no reports of correlations with it. In this study, the public protein data for five subspecies of *Arthrospira* were analyzed to search for a potential c-di-GMP network. A total of 149 potential c-di-GMP turnover proteins were predicted, of which 45% were GGDEF domain proteins (Figure 1 and Appendix A). These proteins are distributed unevenly in each strain; some are homologous, with more than 80% sequence similarity, but some are species-specific (Figure 1 and Appendix A). It can be hypothesized that some c-di-GMP metabolic genes usually function as conserved genes, whereas other specifically distributed genes may function in certain fixed pathways.

To further assess the c-di-GMP network of *Arthrospira*, the conserved protein motifs of each predicted turnover protein were analyzed. In general, active DGCs tend to contain GG[D/E]EF-conserved motifs, whereas PDEs contain E[A/V]L or HD-GYP motifs [5,11]. However, more than 30% of the turnover proteins in each species contain these degenerate domains, and in particular, more than 40% in each of *A. platensis* NIES-39, *A. platensis* C1, and *A. maxima* CS-328, suggesting that these proteins function as c-di-GMP receptors during signaling (Appendix A). Only two EAL domain proteins with conserved motifs (H1WLM6 and B5VU82) were predicted, located in *A.* sp. PCC8005 and *A. maxima* CS-328, respectively (Appendix A). Notably, a protein with a conserved HD-GYP domain was found in all selected strains (Appendix A). It was hypothesized that these proteins containing conserved motifs of EAL or HD-GYP are PDEs. In addition to these conserved motifs, the N-terminal sensory domains such as the REC domain are also critical in regulating the catalytic activity of DGCs and PDEs. The major signal-receiving domains in *Arthrospira* include REC, GAF, PAS, and CHASE2, which are mainly distributed on GGDEF and GGDEF-EAL domain proteins. Some of these may regulate c-di-GMP levels in response to environmental stimuli. In summary, redundant c-di-GMP turnover proteins may predict the existence of a complex c-di-GMP regulatory network in *Arthrospira*.

### 2.2. The GGDEF Domain Protein Adc11 Is an Active DGC

To confirm the presence of an active c-di-GMP network in *Arthrospira*, a predicted PleD-like protein (Adc11) was evaluated, as it was present in all five selected *Arthrospira* strains. This protein contains REC, PAS, PAC, and GAF domains in addition to the GGDEF domain (Figure 2a). Notably, proteins identical to Adc11 are found in most of the *Arthrospira* species deposited in the NCBI database. The genomic context of Adc11 among all sequenced *Arthrospira* genomes in GenBank was compared, and it was found that across almost all typically selected *Arthrospira* species, the genomic context of Adc11 was well conserved (Appendix A). The alignment of all five Adc11 with previously functionally characterized GGDEF domain-containing proteins indicated that the Adc11 possesses a conserved RxGGDEF motif, a c-di-GMP binding RxxD inhibitory (I)-site motif, and a GTP binding (KxxxD) site (Figure 2a). In addition, other characteristic conserved signature amino acids, including the Mg^2+^ binding amino acid, are also present (Figure 2a). These results may suggest that Adc11 is a functional DGC.

To further evaluate the catalytic activity of Adc11, a VC1 riboswitch-based system was constructed for the detection of c-di-GMP concentrations in *Escherichia coli* strain TOP10 in vivo (Figure 2b). In this system, the riboswitch VC1 was engineered as the 5′-UTR of the *lacZ*, allowing for monitoring of c-di-GMP concentration by differential β-galactosidase levels. The VC1 riboswitch has been characterized as an “off” switch in *E. coli* TOP10 [5]. Thus, high intracellular levels of c-di-GMP will bind to VC1 and inhibit the translation of *lacZ*, reducing the hydrolysis of the substrate X-gal and leading to a change in colony color from oxidative blue dye precipitation to white. As the concentration of L-arabinose increased, it was observed that the colonies gradually changed from bright blue to light, indicating that the expression of the five Adc11 significantly increased the concentration of intracellular c-di-GMP and thus inhibited the production of β-galactosidase (Figure 2b). Similar results were observed with the c-di-GMP enzyme-linked immunosorbent assay (Appendix A). In addition, a significant cytotoxicity of Adc11_H1WDI7 overexpression was observed upon 0.01% L-arabinose induction (Figure 2b). Interestingly, this phenomenon has also been shown in some previous studies [5,12,13], suggesting a unique mechanism of action by Adc11_H1WDI7. The fact that high levels of c-di-GMP significantly reduced bacterial motility has been confirmed [14,15]. Therefore, we also attempted to further confirm the catalytic activity of the five Adc11 via swimming assays. As shown in Figure 2c, the swimming ability of *E. coli* harboring Adc11 was significantly down-regulated compared with *E. coli* that are transformed with the empty vector, suggesting the transition from motility to sessility; this process often contributes to multicellular behavior such as biofilm formation [9,16]. Taken together, all these results provide evidence for the presence of an active c-di-GMP network in *Arthrospira*.

### 2.3. The HD-GYP Domain Protein Ahd1 Is Potentially PDE Active

We next evaluated a potential PDE, Ahd1, which, like Adc11, is widely distributed in various *Arthrospira* species. Ahd1 is the only protein with a conserved HD-GYP motif among the five *Arthrospira* strains, comprising an HD-GYP domain fused to a GAF domain (Figure 3a). Compared with GGDEF and EAL, the HD-GYP domain protein has been less characterized because this domain is not encoded in some model organisms, such as *E. coli*. The alignment result of the HD-GYP domains of Ahd1 with some of the most well-characterized HD-GYP proteins, such as CGAP1 (*Vibrio cholerae*) and RpfG (*Xanthomonas campestris* pv. campestris), revealed that Ahd1 has conserved metal ligands, catalytic residues, substrate ligands, and GYP motif (Figure 3a). The genomic content analysis showed that *ahd1* and a molybdopterin precursor biosynthesis protein gene (*moaB*) are located in one cluster. The *ahd1* is invariantly present in the upstream of *moaB* in representative isolates of *Arthrospira* species such as *A. platensis* NIES-39, *A. platensis* C1, and *A.* sp. PCC8005 (Figure 3b). MoaB catalyzes the synthesis of the molybdopterin precursor using guanylic acid derivatives as substrates, which suggests the proximity of substrate synthesis with the molybdopterin precursor synthase to readily produce the molybdenum cofactor (Figure 3c).

We next sought to evaluate the catalytic activity of Ahd1 by swimming motility in *E. coli*. Here, two Ahd1, D5A586 and K1W350, were selected for swimming assays using *E. coli* transformed with an empty plasmid and a validated DGC (Adc11_K1W5H6) as controls. Consistently, *E. coli* harboring Ahd1_D5A586 and Ahd1_K1W350 formed a larger swimming zone than controls, and overexpression of Adc11_K1W5H6 inhibited the swimming motility (Figure 3d). Accordingly, low levels of c-di-GMP were also detected in the two *E. coli* strains expressing Ahd1, whereas overexpression of Adc11_K1W5H6 resulted in high intracellular c-di-GMP (Figure 3d). In summary, the suppression of motility by Ahd1_D5A586 and Ahd1_K1W350, again, added supporting evidence for Ahd1 being catalytically active as a PDE in vivo.

### 2.4. Meta-Analyses of Transcriptomes Suggest That Adc11 and Ahd1 Are Invariable

As discussed above, the distribution of *adc11* and *ahd1* in *Arthrospira* species is conserved. Thus, we hypothesized that Adc11 and Ahd1 function as conserved turnover proteins to adjust a cellular c-di-GMP in *Arthrospira*. To test this hypothesis, a meta-analysis of *adc11* and *ahd1* transcription from the GEO database was performed. After quality control and normalization of the published data sets, a total of 22 different conditions for *Arthrospira* studies were collected, mainly including different growth media and different intensities and durations of radiation. However, the transcription level of *adc11* and *ahd1* showed no significant changes under all experimental conditions (*p* > 0.05, Figure 4a and Appendix A). To further verify the transcriptional pattern of *adc11* and *ahd1*, we tested the gene expression levels of *Arthrospira* under different physiological conditions, such as nitrogen (N) deprivation, different temperatures, and different light intensities, using quantitative real-time PCR. The results showed that the expression of *adc11* and *ahd1* did not significantly change in each condition compared with controls (consistent with our meta-analysis of GEO data sets), which may indicate that their expression was unaffected by various physiological conditions (Figure 4b,c). These data further suggested that Adc11 and Ahd1 are invariable c-di-GMP turnover proteins, implying that their presence in *Arthrospira* may be critical for maintaining stable intracellular c-di-GMP concentrations.

## 3. Discussion

The c-di-GMP signaling network of the filamentous cyanobacterium *Arthrospira* is poorly understood. Here, a total of 149 c-di-GMP turnover proteins was predicted from five *Arthrospira* subspecies. We investigated the phylogenetic distribution of these proteins and noticed that several of them are highly conserved, but some are relatively less conserved or even specifically found only in certain species. These results raised the possibility that specifically distributed genes may function in certain fixed pathways [17]. In this study, we focused on two genes, *adc11* and *ahd1*. They are conserved in the genome and were shown to have DGC or PDE activity, respectively. Notably, the expression pattern and expression level analysis showed a striking feature of *adc11* and *ahd1*, i.e., that their expression may not be affected by physiological conditions. This result supports the possibility that they may act as basic c-di-GMP-producing or -degrading genes in *Arthrospira* to maintain normal levels of c-di-GMP. However, despite a novel DGC and a novel PDE from these *Arthrospira* species that were initially experimentally characterized, the physiological roles of the two proteins are still undefined. In particular, a redundant c-di-GMP network exists in *Arthrospira,* and their biological functions still need to be resolved by deleting these genes individually in *Arthrospira*. In addition, the application of phylogenetic tools or the use of histological datasets to aid in the functional analysis of these genes is essential.

It is well known that *Arthrospira* undergoes morphological variation in response to changes in environmental conditions (e.g., pH, temperature, and light) in both natural and artificial environments [8,18]. Several previous studies of multicellular bacteria have demonstrated the important role of the c-di-GMP network in regulating changes in cell morphology. For example, the inactivation of a DGC (All2874) in *Anabaena* sp. PCC 7120 resulted in an increase in the intervals between the heterocysts from 25 to approximately 200 vegetative cells [2,4]. In contrast, at least four c-di-GMP turnover proteins capable of affecting cell morphology have been reported in *S. coelicolor*, including two DGCs and two PDEs [19,20,21]. Notably, some reports have confirmed that c-di-GMP exerts a global control on natural product biosynthesis in streptomycetes [22,23]. Therefore, it is reasonable to speculate that the c-di-GMP network may also influence morphological variation or the production of their commercially important secondary metabolites in *Arthrospira*. In particular, several studies have shown that the maintenance of the linear morphology of *Arthrospira* is related to the level of peptidoglycan [8], the hydrolysis of which can be regulated by c-di-GMP [24]. However, comprehensive phylogenetic and expression analyses of c-di-GMP-metabolizing genes in *Arthrospira* are required to test this hypothesis. Although the present study did not provide evidence for the specific biological functions of Adc11 and Ahd1 in *Arthrospira*, the bioinformatics and experimental evidence we collected confirmed that *Arthrospira* species possess a functional c-di-GMP signaling network. Overall, this work is expected to support further comprehensive in-depth studies of this system and contribute to a better understanding of the cyanobacterial c-di-GMP network.

## 4. Materials and Methods

### 4.1. Data Analysis

#### 4.1.1. Protein Analysis

Protein data for five *Arthrospira* strains, including *A. platensis* C1, *A. platensis* NIES-39, *A.* sp. PCC8005, *A.* sp. TJSD091, and *A. maxima* CS-328, were obtained from the UniProt databases (https://www.uniprot.org/ (accessed on 23 June 2023)). Protein sequences of predicted DGCs and PDEs were used for domain prediction; this process was performed using the SMART online tool (http://smart.embl-heidelberg.de/ (accessed on 28 June 2023)) as described by Letunic et al. [25]. Protein sequence alignments of Adc11 and Ahd1 were performed using ClustalW and the alignment results were then optimized using ESPript 3.0 (https://espript.ibcp.fr/ESPript/cgi-bin/ESPript.cgi (accessed on 15 July 2023)) [5]. The crystal structure models of PLED_CAUVN and CGAP1_VIBCH were downloaded from PDB databases (https://www.rcsb.org/ (accessed on 6 July 2023)) and used as references for the determination of the secondary structure according to the description by Robert and Gouet [26]. Genomic content was analyzed using the SyntTax online tool (https://archaea.i2bc.paris-saclay.fr/SyntTax/ (accessed on 19 July 2023)). The raw data produced by SyntTax in “.pdf” format was edited for clarity, and genes corresponding to the query proteins were boxed with light color.

#### 4.1.2. Meta-Analysis of GEO Data Sets

Meta-analysis was performed using custom-made R scripts. Briefly, raw datasets used in this study were obtained from the publicly available expression profiling data sets on *Arthrospira* in the GEO database (https://www.ncbi.nlm.nih.gov/gds (accessed on 12 August 2023)). After data filtering, four usable GSE datasets were collected, including GSE175921, GSE67839, GSE63250, and GSE57456. MAS5 normalization implemented in the “Affy” package was used to perform the normalization to obtain expression values for all 8996 probe sets. Then, meta-analysis of the obtained expression profiles was performed, in particular for genes of interest, *adc11* and *ahd1*.

### 4.2. Experimental Methods

#### 4.2.1. Bacterial Strain, Growth Conditions, and Plasmid Construction

*E. coli* strain TOP10 was purchased from Sangon Biotech (Shanghai, China). Cells were cultured in Luria–Bertani (LB) medium with shaking at 200 rpm or grown on LB agar plates at 37 °C. *A.* sp. HN8 was cultured in Zarrouk medium with the growth conditions as described by Milia et al. [27].

For the construction of the translational reporter vector, the VC1 riboswitch was amplified from *Vibrio cholerae* comprising from −240 to +15 bp concerning the open reading frame; the *lac* promoter containing *lac* operator was amplified from pUC19; the N-terminal of *lacZ* was amplified from pRS426; and the *cat* terminator was amplified from pACYCDuet-1. These fragments were fused via overlap extension PCR, resulting in a *lacZ* expression cassette (Appendix A). Then, the fragment of the *lacZ* expression cassette was ligated into pBad18 using a ClonExpress II One Step Cloning Kit (Vazyme, Nanjing, China), resulting in pBad-VC1-*lacZ*. To assess the putative protein’s activity, the gene sequences of predicted proteins with DGC activity were synthesized by the Ruibo company (Qingdao, China) and were subsequently assembled into the pBad-VC1-*lacZ*. All plasmids harboring a predicted DGC and a *lacZ* expression cassette were transformed into chemically competent *E. coli* TOP10 cells for evaluation of c-di-GMP synthesis. All primers are summarized in Appendix A, and *E. coli* strains transformed with plasmids are summarized in Appendix A.

#### 4.2.2. Assessment of C-di-GMP Synthesis Based on VC1 Riboswitch

Transformed *E. coli* TOP10 strains harboring predicted DGCs were inoculated in LB medium supplemented with 100 μg mL^−1^ kanamycin with 200 rpm shaking for 12 h at 37 °C. Subsequently, cultures were diluted to an OD_600_ of 0.1 and continued to be cultured to an OD_600_ of 0.6. Approximately 3 μL of each culture was spotted onto an LB agar plate containing 100 μg mL^−1^ kanamycin, 40 μg mL^−1^ X-gal (Solarbio, Beijing, China), and 0−0.01% (*w*/*v*) L-arabinose (Solarbio). The agar plate was incubated at 30 °C for color development for up to 24 h.

#### 4.2.3. Swimming Motility

The motility of *E. coli* TOP10 derivatives carrying either the empty vector pBad-VC1-*lacZ* or the overproducing predicted DGCs or PDEs were tested in swimming motility *assays.* Strains were grown overnight in LB agar plates (1.5%) supplemented with kanamycin at 37 °C. Then, strains were inoculated in the center of LB agar plates (0.3%) supplemented with kanamycin. Swimming haloes’ diameters were detected using Tanon-5200 (Tanon, Shanghai, China) after 24 h incubation at 30 °C.

#### 4.2.4. Detection of Intracellular C-di-GMP Concentration

*E. coli* TOP10 derivatives carrying either the empty vector pBad-VC1-*lacZ* or the overproducing predicted DGCs or PDEs were inoculated in LB medium supplemented with 100 μg mL^−1^ kanamycin until OD_600_ of 0.6. Two milliliters of each culture were centrifuged at 10,000 rpm for 2 min at 4 °C. The cell pellet was resuspended in 2 mL ice-cold PBS and incubated at 100 °C for 5 min. Then, the samples were sonicated for 10 min (100 W, 4 s/10 s cycle) in an ice-water bath. After centrifugation, the supernatants containing extracted c-di-GMP were collected and determined to be 2 mL. Intracellular c-di-GMP levels were measured using a c-di-GMP enzyme-linked immunosorbent assay (ELISA) kit (Lmai, Shanghai, China). The total protein content was determined using a bicinchoninic acid (BCA) assay kit (Solarbio), and the c-di-GMP level was expressed as pmol mg^−1^ of protein.

#### 4.2.5. Sample Preparation, RNA Extraction, and Quantitative PCR

*A*. sp. HN8 grown to log phase was harvested and washed twice with sterile water. Cells were inoculated and induced under the following conditions: (1) N limitation, N-free Zarrouk medium; (2) temperature, 20, 25, and 35 °C; and (3) light intensity, 1000, 2000, 4000, and 5000 lux. Cells were incubated for 2 h in each condition and harvested for RNA extraction. RNA extraction was performed using the method described by Wang et al. [28]. RNA samples (0.8~1 μg) were used for first-strand cDNA synthesis using *Evo M-MLV* RT Premix for qPCR AG11706 (Accurate Biology, Changsha, China). The cDNA concentrations were then determined using 7500 Fast real-time PCR system (Thermo Fisher Scientific, Waltham, MA, USA) and SYBR Green I detection reagent. The 16S rRNA was used as reference for normalization of real-time PCR data. The relative expression levels of genes were analyzed using the 2^−ΔΔCT^ method.

## 5. Conclusions

In this study, the public protein data for five *Arthrospira* subspecies were analyzed to search for a potential c-di-GMP network in *Arthrospira*. Bioinformatic results showed that at least 149 candidate c-di-GMP metabolizing proteins were located in five *Arthrospira* subspecies. Phylogenetic analysis revealed that some of these proteins are highly conserved, but some are relatively less conserved or even specifically found only in certain species. We focused on two conserved proteins, Adc11 and Ahd1. The former was shown to exhibit DGC activity using a riboswitch-based c-di-GMP assay system; in contrast, Ahd1 was found to significantly reduce the intracellular c-di-GMP concentration of *E. coli*. Notably, both meta-analyses of the transcriptome and qPCR validation experiments showed that Adc11 and Ahd1 are invariable in various physiological conditions, suggesting that they may play a role in maintaining the normal physiological activities of *Arthrospira* as DGCs and PDEs required for the regulation of intracellular basal c-di-GMP levels. In conclusion, the present study demonstrates for the first time the possible existence of an active c-di-GMP network in *Arthrospira*, which will provide support for further studies on the physiological functions of c-di-GMP signaling in *Arthrospira*.

## Figures and Tables

**Figure 1 ijms-24-15210-f001:**
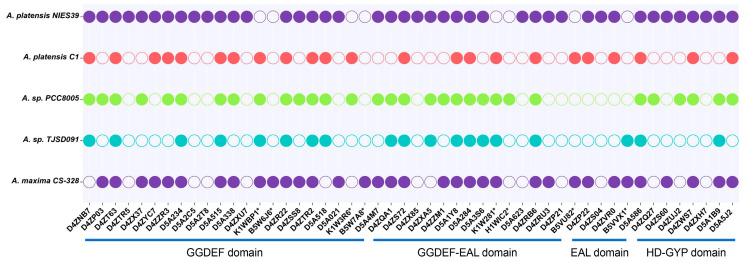
Phylogenetic distribution of predicted DGCs and PDEs in five *Arthrospira* subspecies. The protein data were obtained from UniProt databases. Proteins from *A. platensis* NIES-39 were used as a control. The presence of an ortholog for each predicted DGC among *Arthrospira* subspecies is indicated by a filled circle, and their absence is indicated by an open circle. * means that the protein is absent in *A. platensis* NIES-39, but is present in other subspecies.

**Figure 2 ijms-24-15210-f002:**
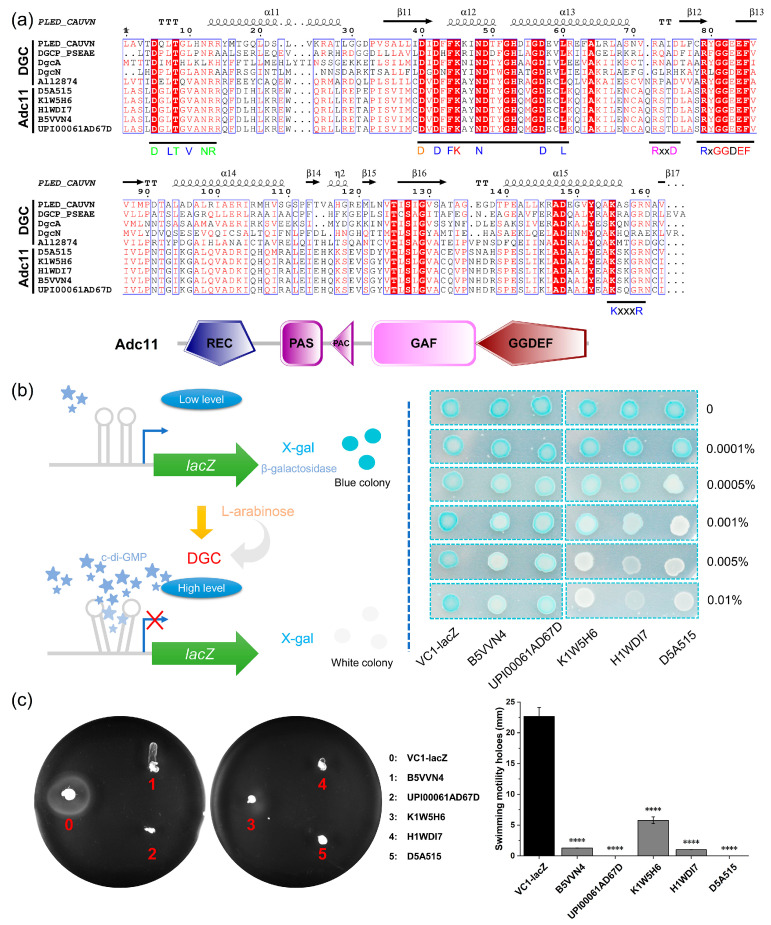
Characterization of the activity of a GGDEF domain-containing protein, Adc11, in *E. coli* TOP 10. (**a**) Alignment of the GGDEF domains of Adc11 with experimentally verified GGDEF domains (PLED_CAUVN, *Caulobacter vibrioides*; DGCP_PSEAE, *Pseudomonas aeruginosa*; DgcA, *Treponema denticola*; DgcN, *E. coli* strain K12; All2874, *Anabaena* sp. PCC 7120). The determination of the secondary structure is based on the PLED_CAUVN crystal structure. I-site, GG[D/E]EF motif, and GTP/Mg^2+^ binding are highlighted in pink, red, and blue, respectively. (**b**) Detection of alterations in c-di-GMP levels via Vc1 riboswitch for candidate DGC Adc11 in *E. coli* TOP10. (**c**) Swimming motility assay of *E. coli* TOP10 derivatives carrying either the empty vector pBad-VC1-lacZ or the overproducing predicted DGC Adc11 at 24 h in LB medium. The results represent the average ± SD of biological triplicates (*n* = 3). Asterisks denote statistical significance of the data according to *t*-tests with Bonferroni–Dunn correction: **** *p* < 0.0001.

**Figure 3 ijms-24-15210-f003:**
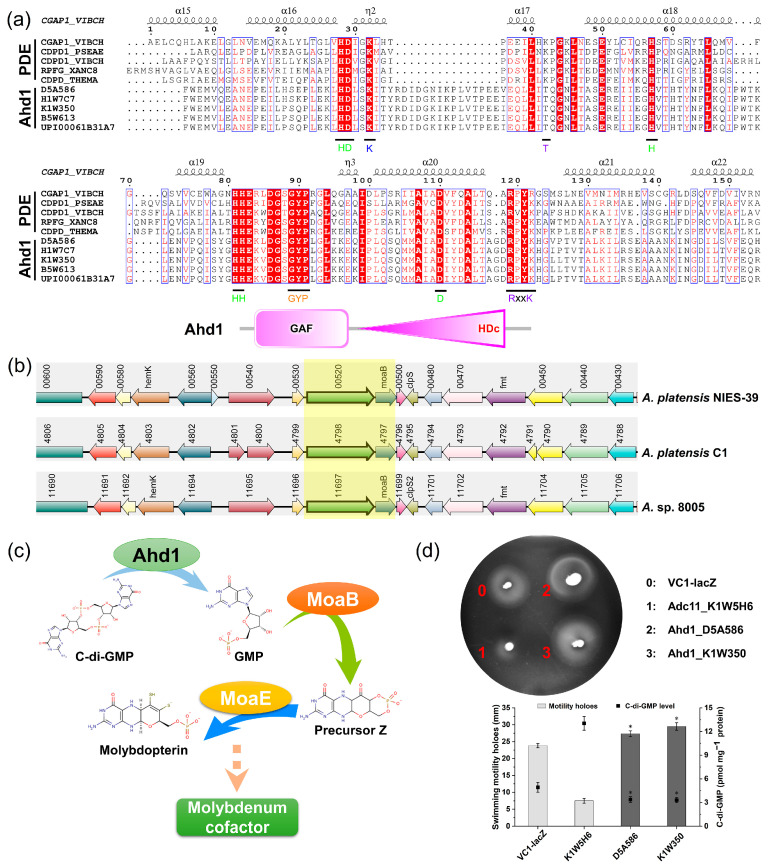
Characterization of the activity of an HD-GYP domain-containing protein, Ahd1, in *E. coli* TOP 10. (**a**) Alignment of the HD-GYP domains of Ahd1 with experimentally verified HD-GYP domain-containing proteins (CGAP1_VIBCH and CDPD1_VIBCH, *Vibrio cholerae* serotype O1; CDPD1_PSEAE, *Pseudomonas aeruginosa*; RPFG_XANC8, *Xanthomonas campestris* pv.; CDPD_THEMA, *Thermotoga maritima*). The determination of the secondary structure is based on the CDPD1_PSEAE crystal structure. Metal ligands, catalytic residues, substrate ligands, and GYP motif are highlighted in green, blue, violet, and orange, respectively. (**b**) Synteny analysis of Ahd1. Complex syntenies were obtained using the protein sequence of *A. platensis* C1 Ahc1_K1W350 as a query sequence to search against the indicated *Arthrospira* subspecies. Genes corresponding to the query proteins are boxed with light yellow. (**c**) Diagram of the possible role of Ahd1 in *Arthrospira*, i.e., to provide substrate for molybdenum cofactor synthesis by degrading c-di-GMP. (**d**) Swimming motility assay and detection of intracellular c-di-GMP concentration of *E. coli* TOP10 derivatives carrying either the empty vector pBad-VC1-lacZ or the overproducing predicted DGCs and PDEs at 24 h in LB medium. The results represent the average ± SD of biological triplicates (*n* = 3). Asterisks denote the statistical significance of the data according to *t*-tests with Bonferroni–Dunn correction: * *p* < 0.05.

**Figure 4 ijms-24-15210-f004:**
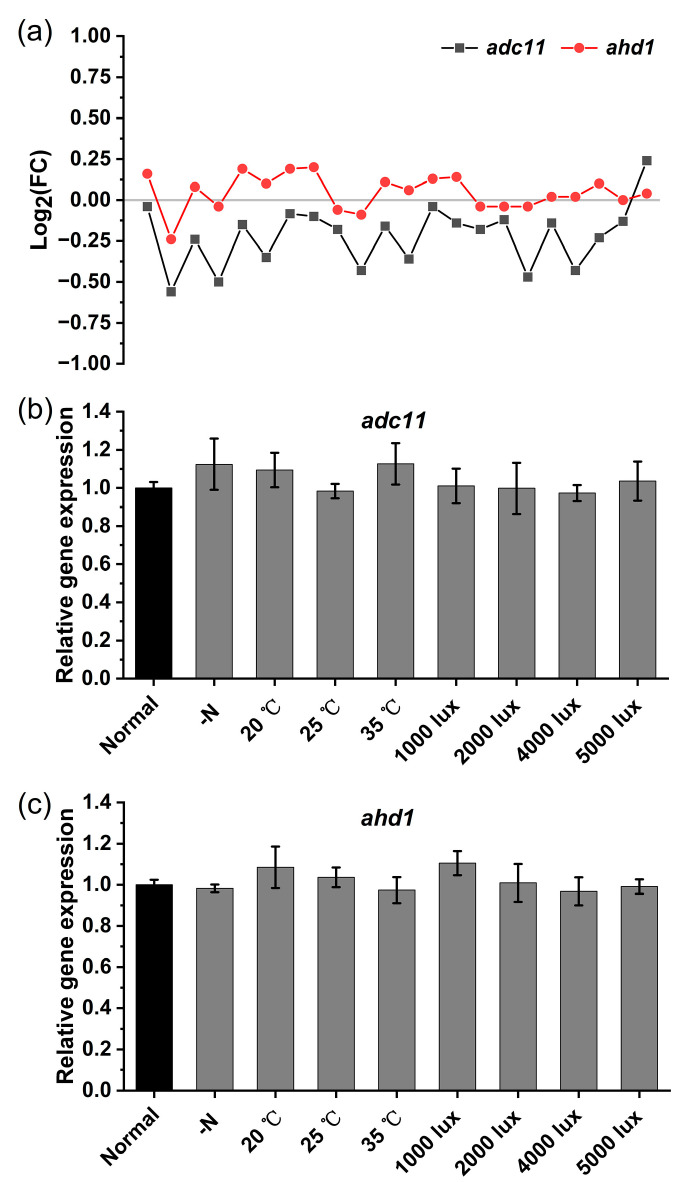
Meta-analyses of transcriptomes suggest that Adc11 and Ahd1 are invariable proteins in *Arthrospira*. (**a**) Relative expression profiling patterns of *adc11* and *ahd1* in *Arthrospira* meta-array analyses. A total of 22 conditions were tested for expression analyses of *adc11* and *ahd1*, and no significant differences were observed. (**b**) *adc11* expression levels under different environmental conditions. (**c**) *ahd1* expression levels under different environmental conditions. For each condition, the relative expression levels were obtained by comparing gene expressions under normal growth conditions.

## Data Availability

The authors affirm that all the data necessary for confirming the conclusions of the article are present within the article, figures, and tables.

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
