# Peer review of "Phylogenetic Analysis and Characterization of Diguanylate Cyclase and Phosphodiesterase in Planktonic Filamentous Cyanobacterium Arthrospira sp."

_ijms, 2023, doi:10.3390/ijms242015210_

Round 1
Reviewer 1 Report
Please see attached

Minor editing may be required
Author Response
Dear Reviewer,
Thanks for your valuable suggestions for our manuscript. Please find attached the revised manuscript using a RED font of our paper entitled "Phylogenetic analysis and characterization of diguanylate cyclase and phosphodiesterase in planktonic filamentous cyanobacterium Arthrospira sp." together with a response to your comments.
With kind regards.
- Line 106-107 - did authors check for the location of characteristic binding sites and amino acid residues that were identified through comparison of amino acid sequences, using additionally an in silico 3D protein homology modeling and structural alignment of obtained models with the known crystal structures of similar proteins?
Response: Thanks for your suggestion. In fact, the 3D modeling of both Adc11 and Ahd1 has been established using the published modeling of DGC and PDE, respectively. Then, the pairwise sequence alignment between the known crystal structure and Adc11/Ahd1 was performed using the DALI protein structure comparison server. Despite differences in the N-terminal structural domains of the proteins, the results showed that their positions as functional sites of the DGC/PDE are identical. However, given the length of the article and the highly conserved amino acid sequences in characteristic binding sites, these results are not listed and discussed in the manuscript.
- Line 207 - suggest to clarify which nitrogen concentration was used in the experiment as limiting and which nitrogen source was used for the test.
Response: Thanks for your suggestion. The information on nitrogen concentration has been added in the Experiment Methods section, please check it. Moreover, the description of nitrogen concentration in Line 207 was changed to "…nitrogen (N) deprivation…".
- Line 237 - does sp. refer to Arthrospira sp.? Please specify.
Response: Thanks for your suggestion. All the descriptions of A. sp. PCC7120 in the manuscript has been changed to "Anabaena sp. PCC7120". Please check it.
- Line 239-241 - suggest to add that besides cell morphology in S. coelicolor there is also an effect on secondary metabolism reported in [21]. May it be important for Arthrospira sp. potentially in regard to the secondary metabolism (if reported).
Response: Thanks for your suggestion. Some discussion has been added, i.e., "Notably, some reports have confirmed that c-di-GMP exerts a global control on natural product biosynthesis in streptomycetes [22, 23]. Therefore, it is reasonable to…".
Reviewer 2 Report
The brief report titled: “Phylogenetic analysis and characterization of diguanylate
cyclase and phosphodiesterase in planktonic filamentous cyanobacterium Arthrospira sp.”
is an interesting report from the point of view of the methodology used in the work.
However, a few comments arise:
1. Fig. 1. The side descriptions of the drawings are illegible
2. Fig. 2 The capitalization of letters in points a, b and c requires correction
3. Fig. 3 Illegible descriptions in points a, b and c require correction
4. Line 61-62. The sentence needs to be reworded.
5. Line 208 The transcription of adc11 and ahd1 did not significantly change compared to controls (consistent with our meta-analysis of GEO data sets), indicating that their expression was unaffected by various physiological conditions (Fig 4b and 4c).This needs to be made more precise.
6. The chapter Materials and Methods particularly Line 263-270 should be expanded.
7. The conclusions need to be rewritten, the conclusions are too general. Please expand and specify your statement.
Author Response
Dear Reviewer,
Thanks for your valuable suggestions for our manuscript. Please find attached the revised manuscript using the RED font of our paper entitled "Phylogenetic analysis and characterization of diguanylate cyclase and phosphodiesterase in planktonic filamentous cyanobacterium Arthrospira sp." together with the response to your comments.
With kind regards.
- Fig 1. The side descriptions of the drawings are illegible
Response: Thanks for your suggestion. The description of Fig. 1 has been improved and changed to "Phylogenetic distribution of predicted DGCs and PDEs in five Arthrospira subspecies. The protein data was obtained from UniProt databases. Proteins from A. platensis NIES-39 were used as a control. The presence of an ortholog for each predicted DGC among Arthrospira subspecies is indicated by a filled circle, and their absence is indicated by an open circle. * means that the protein is absent in A. platensis NIES-39, but is present in other subspecies".
- Fig 2. The capitalization of letters in points a, b and c requires correction
Response: Thanks for your suggestion. We have fully checked the capitalization of letters in Fig. 2. In fact, the capitalization of all letters in Fig. 2 is accurate.
- Fig. 3 Illegible descriptions in points a, b and c require correction
Response: Thanks for your suggestion. We have fully checked it and added some new descriptions in Fig. 3, such as "a) Alignment of the HD-GYP domains of Ahd1 with experimentally verified HD-GYP domain-containing proteins" and "Genes corresponding to the query proteins are indicated in boldface and boxed with light yellow".
- Line 61-62. The sentence needs to be reworded.
Response: Thanks for your suggestion. The sentence was changed to "In 1998, two linear DNA fragments were isolated from A. platensis FS by Kojima et al. [10]. These fragments are homologous to the A. platensis genome and bioinformatic results suggested…".
- Line 208 The transcription of adc11 and ahd1 did not significantly change compared to controls (consistent with our meta-analysis of GEO data sets), indicating that their expression was unaffected by various physiological conditions (Fig 4b and 4c). This needs to be made more precise.
Response: Thanks for your suggestion. The sentence has been changed to "The results showed the expression of adc11 and ahd1 did not significantly change in each condition compared to controls (consistent with our meta-analysis of GEO data sets), which may indicate that their expression was unaffected by various physiological conditions".
- The chapter Materials and Methods particularly Line 263-270 should be expanded.
Response: Thanks for your suggestion. Some details were added, such as "Protein sequences were used for domain prediction; this process was performed using the SMART online tool (http://smart.embl-heidelberg.de/) according to the description by Le-tunic et al. [24]", "…used as references for the determination of the secondary structure according to the description by Robert and Gouet [25]" and "The raw data produced by SyntTax in “.pdf” format was edited for clarity and genes corresponding to the query proteins were boxed with light color". Please check it. Moreover, we have added relevant references to allow the reader to repeat these results.
- The conclusions need to be rewritten, the conclusions are too general. Please expand and specify your statement.
Response: Thanks for your suggestion. The conclusion has been rewritten, i.e., "In this study, the public protein data for five Arthrospira subspecies were analyzed to search for a potential c-di-GMP network in Arthrospira. Bioinformatic results showed that at least 149 candidate c-di-GMP metabolizing proteins were located in five Arthrospira subspecies. Phylogenetic analysis revealed that some of these proteins are highly conserved, but some are relatively less conserved or even specifically found only in certain species. We focused on two conserved proteins, Adc11 and Ahd1. The former was shown to have DGC activity using a riboswitch-based c-di-GMP assay system; in contrast, Ahd1 was found to significantly reduce the intracellular c-di-GMP concentration of E. coli. Notably, both meta-analyses of the transcriptome and qPCR validation experiments showed that Adc11 and Ahd1 are invariable in various physiological conditions, suggesting that they may play a role in maintaining the normal physiological activities of Arthrospira as DGC and PDE required for the regulation of intracellular basal c-di-GMP levels. In conclusion, the present study demonstrates for the first time the possible existence of an active c-di-GMP network in Arthrospira, which will provide support for further studies on the physiological functions of c-di-GMP signaling in Arthrospira".